# Enhanced Magnetic Behavior of Cobalt Nano-Rods Elaborated by the Polyol Process Assisted with an External Magnetic Field

**DOI:** 10.3390/nano10020334

**Published:** 2020-02-15

**Authors:** Mohamed Ali Bousnina, Amel Dakhlaoui-Omrani, Frédéric Schoenstein, Yaghoub Soumare, Aliou Hamady Barry, Jean-Yves Piquemal, Guillaume Viau, Silvana Mercone, Noureddine Jouini

**Affiliations:** 1Laboratoire des Sciences des Procédés et des Matériaux, CNRS, UPR 3407, Université Sorbonne Paris Nord, 99 Avenue J.B. Clément, F-93430 Villetaneuse, France; medalibousnina@yahoo.fr (M.A.B.); amel_dakhlaoui@yahoo.fr (A.D.-O.); frederic.schoenstein@univ-paris13.fr (F.S.); soumareyaghoub@yahoo.fr (Y.S.); barryaliouhamady@ymail.com (A.H.B.); 2Department of Chemistry, Faculty of Sciences and Arts-Khulais, University of Jeddah, Khulais P.O. Box 355, Jeddah 21921, Saudi Arabia; 3Laboratoire de Chimie des Matériaux, Département de Chimie, Faculté des Sciences et Techniques, Université des Sciences de Technologie et de Médecine, Nouakchott BP 880, Mauritanie; 4Laboratoire Interfaces Traitements Organisation et Dynamique des Systèmes, CNRS, UMR 7086, Université de Paris, 15 rue J.-A. de Baïf, F-75013 Paris, France; jean-yves.piquemal@u-paris.fr; 5Laboratoire de Physique et Chimie des Nano-objets, CNRS, INSA, UPS, Université de Toulouse, 135 avenue de Rangueil, F-31077 Toulouse CEDEX 4, France; guillaume.viau@insa-toulouse.fr

**Keywords:** cobalt, nano-rods, polyol process, magnetic field, ferromagnetism

## Abstract

Cobalt nano-rods with the hexagonal close-packed (hcp) structure were prepared by reduction of the long-chain carboxylate Co (II) precursor in polyol. The application of an external magnetic field (*µ*_0_*H* = 1.25 T) during the nucleation and growth steps resulted in a noticeable modification of the mean aspect ratio (length/diameter) of the particles. The particle morphology was also modified as the nano-rods did not exhibit conical heads at their extremities anymore, which are observed for particles prepared without application of an external magnetic field. Besides, the stacking faults density along the *c* axis of the hcp structure in the cobalt nano-rods has been found to decrease with the increase in the applied magnetic field. The coercive field of randomly oriented nano-rods increased with the aspect ratio, showing the highest value (i.e., 5.8 kOe at 300 K) for the cobalt nano-rods obtained under the highest applied magnetic field. For partially oriented Co nano-rods in toluene solution, the magnetic properties were significantly enhanced with a coercive field of 7.2 kOe at 140 K, while the magnetization saturation reached 92% of the bulk. The *M_R_/M_S_* value was about 0.8, indicating a good orientation of the anisotropic particles relative to each other, making them suitable for the preparation of permanent magnets via a bottom-up approach.

## 1. Introduction

Magnetic materials present several interests in the field of high-density magnetic recording [1], spintronics, and microwave applications [2], and can be used as new building blocks for the preparation of permanent magnets [3]. The major interest of ferromagnetic metals with respect to oxides is their higher saturation magnetization. Furthermore, elongated single-domain particles present a high magnetic anisotropy. The combination of high anisotropy with high saturation magnetization makes these kinds of particles very attractive as precursors for hard magnetic materials. Furthermore, a high aspect ratio, i.e., the ratio of the mean length to the mean diameter, likely opens new perspectives in areas such as the measurement of the properties of electrical conduction in metallic nanowires. Indeed, the main studied systems are based on nanowires of noble metal nanowires (Ag, Au) [4,5]. Co nanorods with very long length, elaborated in the present study, offer, so far, a new field of investigations for interconnected unidimensional systems that can respond to an external magnetic field stimulus.

The reduction of cobalt carboxylates in a liquid polyol at high temperature is a simple way to produce metallic Co nanoparticles. We have recently shown that it is possible to obtain Co nano-rods (NRs) by heterogeneous nucleation with a fine control of the nucleation step and of the growth rate [6,7]. These particles crystallize in the hexagonal close-packed (hcp) phase with the growth axis along the crystallographic *c* axis. The NRs can be considered as nanocrystals even if some stacking faults can be detected by high-resolution (HR)TEM. The main parameters governing the growth are the basicity of the medium, the nature of the Co (II) precursor, the temperature ramping rate, the nature of the polyol, and the stirring rate during the synthesis [6,8].

However, TEM data revealed that these NRs have two conical heads at their ends whose impact on the magnetic behavior of the nano-rods facilitates magnetization reversal and, thus, reduces their hard magnetic properties. Micromagnetic simulations, as well as experimental results, have demonstrated that the conical heads at the ends of the nano-rod particles strongly contribute to the coercivity decrease due to the high demagnetizing field at the conical endings. This is a major issue excluding them from the permanent magnet applications [9,10,11]. In addition, using the standard polyol process, described above, it is very difficult to extend the mean length of the nano-rods above 300 nm.

In this paper, we present results concerning the formation of NRs prepared using a method based on the joint use of the polyol process and the application of an external magnetic field. The resulting particles display higher mean lengths (up to 570 nm) than those obtained without external magnetic fields, while the mean diameter is still in the nanometer range (15 nm). This magnetic field-assisted polyol process also leads to a decrease in the stacking faults along the *c* axis of the hcp structure. Moreover, an optimization of the shape of the Co nanoparticles has also been obtained as the conical heads disappeared at their extremities. In order to optimize the cobalt nanoparticle properties, we have studied their magnetic behavior as a function of the magnetic field applied during their synthesis. Our results show that the NRs obtained under the application of the magnetic field present good hard magnetic properties at room temperature.

## 2. Experimental

### 2.1. Synthesis

CoCl_2_·6H_2_O (Alfa Aesar, 99.9%, Kandel, Germany), sodium laurate (Aldrich, 99.5%, Saint Quentin Fallavier, France), NaOH micro-pearls (Acros, Geel, Belgium), RuCl_3_·xH_2_O (Aldrich, 99.98%), 1,2-butanediol (Aldrich, ≥98%), methanol (VWR, Normapur, Fontenay sous Bois, France), and ethanol (VWR, Normapur) were used without any further purification.

Appendix A shows a schematic drawing of the setup and of the experimental protocol adopted to elaborate the cobalt NRs. This protocol consists of two steps. First, the cobalt (II) dodecanoate (laurate) precursor, Co(C_11_H_23_COO)_2_, is prepared following a procedure adapted from a previously published synthesis for the cobalt hexadecanoate (palmitate) salt [12]. An aqueous solution (5.0 mL) of CoCl_2_·6H_2_O (10.5 mmol, 2.5 g) was added dropwise to an aqueous solution (15 mL) of sodium laurate (27 mmol, 6.0 g) at 333 K with vigorous stirring. A purple precipitate formed, and the mixture was further stirred for 15 min. The precipitate was recovered by filtration, washed three times with 50 mL of methanol, and dried overnight at 323 K in an oven.

Cobalt laurate (Co concentration of 0.080 M) and sodium hydroxide (0.075 M) were dissolved in 50 mL of 1,2-butanediol. After total dissolution of cobalt laurate, a small amount of hydrated ruthenium chloride (RuCl_3_⋅xH_2_O) (the Ru/(Co + Ru) molar ratio is fixed to 2.5%) was added as a heterogeneous nucleating agent. The total mixture was then homogenized by stirring for 10 min.

The synthesis was conducted in two reactor types having the same capacity (70 mL). The first one is an open double-walled glass reactor equipped with a cooler. The second one is closed double-walled made of non-magnetic austenitic steel. The latter is hermetically sealed with a threaded plug to maintain the pressure. In both cases, the heating is ensured by circulation in the space surrounding the inner wall of a heat transfer oil using a thermostatically controlled bath (see Appendix A).

The reactor with the mixture introduced in the inner layer is placed in the air gap of the electromagnet (See Appendix A). The oil bath is then heated to 170 °C with a temperature ramping rate of 8 °C/min. Thanks to a prior magnetic field calibration (See Appendix A), it is possible to vary the applied magnetic field from 0 to 1.25 T by controlling the value of the current intensity applied to the electromagnet. The magnetic field is maintained during both synthesis and cooling steps.

During our investigations, we first carried out the syntheses in an open reactor (OR) under atmospheric pressure (P = 1 atm) and varied the applied magnetic field from 0 to 1.25 T, in order to study the influence of this field on the main characteristics of the synthesized rods. This allowed us to determine the optimum applied field leading to the desired characteristics: Long rod length, small diameter, and absence of heads. In the second step, and by setting the value of the optimal field, we conducted a synthesis in a closed reactor (CR), with all other conditions kept identical except for the autogenous pressure, which was about 50 atm. The aim of this second step was to check the influence of the autogenous pressure on the microstructural characteristics of the NRs.

The samples were denoted Hx-OR (or CR), where x corresponds to the intensity of the applied magnetic field (*µ*_0_*H* in the range 0–1.25 T). After 1 h of reaction at 170 °C, the reduction was complete and the solution was cooled down. In order to preserve the particles from oxidation, they were kept in the polyol solution. The magnetic powder was recovered for physicochemical analyses by centrifugation, washed with absolute ethanol, and dried under vacuum at 323 K.

### 2.2. Characterization

X-ray diffraction (XRD) patterns obtained using Co Kα radiation were recorded on an INEL EQUINOX 1000 diffractometer (INEL, Artenay, France). The lattice parameters and the size of coherent diffraction domains (crystallites), L_hkl_, were calculated using the FullProf program [13] based on the Rietveld method [14].

Transmission electron microscopy observations were carried out with a JEOL-2011 TEM microscope (TEM, JEOL, Tokyo, Japan) operating at an accelerating voltage of 200 kV. The mean length and the mean diameter were determined by statistical analysis by counting at least 200 particles using the Image Tool program.

Magnetic measurements of nano-powders were performed with a Quantum Design MPMS 3 magnetometer (Quantum Design Inc., San Diego, CA, USA). Magnetization as a function of the magnetic field was collected in the magnetic field range of [−2 T, +2 T] at 140 and 300 K.

## 3. Results and Discussion

### 3.1. Shape and Structure Characterizations

For the syntheses conducted in an open reactor (OR), Figure 1 shows that the obtained phases correspond to Co whatever the intensity of the applied magnetic field. The best refinements of these diffractograms were obtained using a hexagonal crystal symmetry with almost similar lattice parameters (*a* = 2.5065 (8) Å, *c* = 4.0701 (13) Å) of the hcp Co phase (JCPDS card N° 01-089-7094). Figure 2 gives the Rietveld refinement of the H0.75-OR sample, which stands for the results obtained for the whole series (all the Rietveld refinements obtained on our samples are reported in Appendix A).

Figure 3 gathers together the TEM images of the as-obtained samples prepared in an open reactor, while their main microstructural characteristics are given in Table 1.

Examination of all the results shows that the application of the magnetic field had a noticeable influence on the main microstructural characteristics. The increase in the applied field allowed the formation of NRs of increasing length (256–538 nm) and smaller diameter (35–19 nm). This resulted in a significant increase in the aspect ratio *L/D*, varying from 6.5 to 28.2.

The TEM images (Figure 3) reveal another interesting result. Whereas the NRs obtained without the application of a magnetic field had conical heads at their ends, similar to the nanoparticles reported in previous works [6], when the applied field increased, the number of NRs not presenting the conical heads statistically increased at the expense of those presenting the conical heads. Figure 4 is a schematic representation of the evolution of the Co nano-rods morphology as a function of the applied magnetic field.

The X-ray diffraction patterns are typical of an anisotropic crystal growth with very narrow (00.2) peaks and significantly broader (10.0) peaks. As shown in Table 1, the crystallite sizes along the *c* axis are significantly higher than those along the *a* axis of the hexagonal cell. It is interesting to note that the diffraction coherent domain size, or crystallite size, along the *c* axis increases when the NR length increases, which resulted from an increase in the applied field. The highest value obtained (34 nm) for OR-type samples corresponds to the cobalt NRs showing the highest length, which have been obtained under 1.25 T. Table 1 also reveals that the crystallite size inferred from X-ray diffraction along the *a* axis is almost similar to the diameter of the NR inferred from the TEM observation, showing the monocrystalline characteristic along this crystallographic direction. Conversely, along the *c* axis, the nano-rod presents a polycrystalline characteristic: The number of crystallites per nano-rod varies from 13 to 20. This polycrystalline characteristic is due to the well-known stacking faults defect occurring along the *c* axis of the Co hcp structure. This results from the breakdown in the hcp “ABABAB..” packing sequence into the fcc “ABCABCABC..” one over few atomic planes [15]. It is interesting to note that the stacking faults density calculated according to [16,17] decreases from 5.21% to 2.96% when the NR length increases from 256 to 538 nm. The reduction in defects likely favors the crystalline order and improves the magnetic properties [18].

Setting the applied magnetic field at its optimum value (1.25 T), the synthesis conducted in a closed reactor (CR) shows that the autogenous pressure induced noticeable changes, as can be seen in the TEM images (Figure 3) and in Table 1: (i) The aspect ratio is significantly increased (37.5) as the nano-rods present a mean length of 563(79) nm, whereas the mean diameter is about 15(4) nm, and (ii) the conical heads at the end of the particles become scarce. The inset in Figure 5 shows well-aligned NRs free of conical ends.

A selected area electron diffraction (SAED) pattern performed on a single Co NR (H1.25-CR) was indexed in the hcp system, showing that the growth axis of the particles was parallel to the *c* axis of the hcp structure (Figure 6).

The X-ray diffraction pattern of this sample (Figure 7) is similar to that obtained in the first set of experiments. This sample shows a well-defined anisotropic shape, presenting a very high aspect ratio (37.5) and a high crystallite size along the *c* axis (79 nm), which led to the narrowest (00.2) peak in comparison to the samples obtained in the same elaboration conditions in an open reactor.

A large number of studies have been devoted to the synthesis of anisotropic cobalt particles using an external magnetic field applied to the synthesis reactor. In most of the studies reported in the literature, cobalt precursors were inorganic salts such as cobalt (II) nitrate, cobalt (II) chloride, or cobalt (II) acetate. In these methods, two main approaches can be distinguished. The first one consists of preparing, as a first step, the metallic particles, and, as a second one, in organizing these nanoparticles thanks to the application of a magnetic field. For instance, Xiong et al. [19] prepared Co nanoparticles 80 nm in size following this process. These nanoparticles can be organized using an applied magnetic field to form regular one-particle-thick chain-like structures with a length higher than one micrometer. The second approach consists of applying an external magnetic field while the reduction of cobalt (II) ions into cobalt is occurring [20,21,22]. The applied field seems to be inefficient at influencing the nucleation and mechanism growth. Indeed, again, it was observed that, first, spherical nanoparticles were obtained, and then the applied field induced the aggregation and coalescence of these nanoparticles along a specified direction, allowing the formation of long polycrystalline chains.

In the present work, cobalt carboxylates with long chains (laurate) were used as a precursor salt instead of a generally used commercially available precursor such as cobalt acetate. Indeed, Soumare et al. [6] showed that, starting from cobalt laurate, the polyol process led to cobalt nano-rods with lengths up to 300 nm and diameters around 20 nm. The anisotropic growth of these nano-rods along the *c* axis of the hcp structure was explained on the basis of the preferential adsorption of laurate species on the different faces of the hcp habit [8,23]. Based on the estimation of the chemical potential of the ligand (laurate), it was established that, in the first step of the crystal growth, the laurate is preferentially adsorbed on the lateral facets. This favors the growth of the nano-rods along the *c* axis. In the second step, the chemical potential of the adsorption on the (00.1) face becomes more important, inducing a crystal growth perpendicular to the *c* axis. This leads to the formation of heads at the both ends of the nano-rod.

In the present work, we show that when the synthesis protocol, starting from cobalt laurate, is assisted by an external magnetic field, significant changes in the morphology of the anisotropic nanoparticles are observed. The length of the NRs increases monotonically with the applied field, reaching a value as high as 560 nm. Indeed, the applied field helps Co atoms, with a high magnetic moment, to deposit along the *c* axis of the nanorods, which is their easy magnetization direction that is supposed to be aligned along the applied field direction. This leads to the formation of longer nanorods with better crystallinity (i.e., less stacking faults) and, thus, better magnetic properties. Such influence has already been reported in several previous works on the synthesis of Co anisotropic particles [24,25,26]. Conversely, the diameter of NRs decreases monotonically with the applied field, leading to a high aspect ratio up to 28. Furthermore, conical heads are almost removed by the field-assisted polyol synthesis. This latter phenomenon seems to be enhanced when the synthesis is conducted in a closed reactor as the aspect ratio increases up to 37. These changes can be related to the effect of the external magnetic field on the overall properties of the liquid medium. Indeed, the external field can affect the surface energy and the mobility of ions, particularly, the paramagnetic ones [27].

Besides the applied magnetic field, which appears to play the key role in the optimization of the morphology of cobalt nano-rods, the autogenous pressure can also help to improve the anisotropic growth of long cylindrical nanoparticles without conical ends. Indeed, the reaction conducted in a closed reactor generates autogenous pressure as a solvothermal method. The latter method has been employed by Gandha et al. [28] to produce Co NRs free of conic heads and with a length similar to that obtained in the present work. However, the role of autogenous pressure in the cited work remains to be discussed; the solvothermal process was conducted in the presence of hexadecylamine (HDA), which acts as a shape-controlling agent with a preferred coordination to the lateral facets.

### 3.2. Magnetic Properties

Cobalt NRs have been creating a huge interest in the community of researchers working on metal nano-materials due to their potential in hard magnetic nanostructured devices operating at room temperature. As previously reported [9,10,11], the magnetic behavior of the ferromagnetic anisotropic nanoparticles is driven by two main contributions: The magnetocrystalline anisotropy constant and shape anisotropy constant. In cobalt-based NRs with the hcp structure, the morphology of the anisotropic particles has a key role on the optimization of their ferromagnetic properties. The mean aspect ratio of the nano-rods [9,10], their stacking faults density [18], as well as the conical heads at the end of the nanoparticles (thickness and width) [10,11], and their magnetic alignments [29] are all parameters playing an influential role on the magnetic static behavior of the nano-rods ensemble.

In order to study the magnetization behavior of the NRs prepared in an open reactor and to show the optimized morphology by the magnetic field-assisted synthesis, we mechanically blocked the nano-rods in a resin without taking any precautions for their organization inside this latter. In this configuration, the nano-powders are, thus, randomly oriented and blocked into the resin. We measured, by a standard magnetometer technique, the magnetization of the blocked nano-powders as a function of the applied magnetic field (hysteresis loops) at room temperature (300 K) from −2 to +2 T. The hysteresis cycles saturated at the highest fields (*µ*_0_*H* > 1.5 T). The saturation magnetization was determined by two different methods: The linear extrapolation and the law of approach to saturation. The two obtained values are very similar, the difference being less than 0.8 emu/g. The zoom of the hysteresis loops is presented in Figure 8; it indicates that all our samples present a hard magnetic behavior at 300 K with a ferromagnetic typical open cycle. We summarize the static magnetic properties of the nano-rods in Table 2.

As it can be seen, the magnetic characteristics improved when the aspect ratio increased. The magnetization at saturation *Ms* increased from 133 to 152.5 emu/g, which represents respectively 80% and 92% of the bulk expected value [8]. The variations in the coercive field *H_C_* as a function of the aspect ratio and as a function of the stacking fault density are given respectively in Figure 9a,b. As it can be seen, *H_C_* significantly improved when the aspect ratio increased and when the stacking fault density decreased, in good agreement with theoretical predictions, as well as previous experimental results obtained on CoNi nano-rods [9,10] and on Co nano-rods elaborated in polyol with various stirring rates [8]. Furthermore, the experimental results in Figure 9a confirm the previous experimental expectation on the saturation of the coercive field for NRs with a mean aspect ratio higher than 15. The energy product (*BH*) maximum has been calculated on the basis of a previously reported method [30,31]. The calculations were made considering the density of pure cobalt (d = 8.92 g/cm^3^) and also based on a 100% volume fraction. The magnetic energy values obtained for the different powders also increase when the aspect ratio increases (Figure 9c) and reach 10.7 MGOe, which is of the same order of magnitude as the energy reported by Ener et al. for consolidated Co nanowires [32]. This energy product can be improved with a higher quality of alignment of the nanowires and their dispersion, as reported by Gandha et al. [28], Anagnostopoulou et al. [30], and Fang et al. [33].

It is remarkable to notice that the NRs obtained during the polyol synthesis under the application of 1.25 T presents high magnetic performances: A high coercive field (5.83 kOe) and a *Ms* approaching the saturation magnetization of the bulk (92% of the cobalt bulk value) at 300 K. Nano-rods elaborated in identical conditions but without the assistance of the applied magnetic field present a coercive value of *H_C_* = 4.5 kOe and a magnetization at saturation *Ms* corresponding to 70% of that of the Co bulk at 300 K, as reported in [6]. Keeping in mind the fact that all our samples are crystallized in a hexagonal structure, and thus present the same magnetocrystalline anisotropy, the variation in the magnetic properties observed here is connected to two main parameters: The shape anisotropy and the stacking faults density. These two parameters, in turn, have been easily optimized by the magnetic field-assisted elaboration.

As shown in Table 2, the ratio *M_R_/M_S_* increases with the aspect ratio *L/D*, indicating that the NRs align parallel to each other spontaneously and better when their length (see Table 1) increases. As reported previously [34,35], the optimized shape anisotropy (*L/D* = 28) induces an easy magnetization axis along the length of the NRs, inducing, at its turn, a high dipolar field outside of the NR. This field is radial around the NR ends and parallel along the lateral surface and decreases quickly as a function of the distance. The dipolar interaction of this magnetic field with the magnetic moment of the nearby NRs tends to align the close NRs parallel to each other. This is more and more effective when the magnetic moment is maximized inside the NRs, and the magnetic anisotropy forces it to align along the length. This is the case in the NRs with optimized aspect ratios (see Table 1 and Table 2). It has also been demonstrated [9,10,11] that the radial dipolar field at the end of the NRs strongly depends on the shape of the NR heads. A conical head at the end of the NR creates a higher demagnetizing field outside this latter, compared to that at the head of the cylindrical NR. This has a catastrophic effect on the spontaneous alignment of the nearby NRs, as well as on their coercivity value. As reported before [35], the magnetization reversal mechanism of NRs ensemble strongly depends on the orientation of few NRs close to the head of the nearby ones. Our measurements in Figure 8 are in full agreement with those previous observations. However, an almost perfect alignment is not obtainable at room temperature in the case of dried nano-powders, because of the spontaneous nano-rods agglomeration during the drying process. To check the influence of this alignment on the magnetic characteristics, we decided to align the nano-rods before the magnetic measurements. In order to do this, we used the particles (H1.25-CR) presenting the best aspect ratio and dispersed them in liquid toluene. A magnetic field of 50 kOe was applied during the freezing of the solution (below 140 K). It is important to underline that this sample is expected to have the same static magnetic characteristics of the corresponding sample synthesized in the same conditions but in an open reactor (H1.25-OR). As described before, the effect of the closed reactor has been studied only in the case of the 1.25 T elaborated sample. The observed effect of the closed reactor compared to the open one is that the presence of the conic heads at the end of the NRs seems to have been significantly reduced in the former case. Thus far, we can assume that NRs with similar structural hexagonal symmetry and comparable mean aspect ratio have a similar magnetic static behavior at room temperature. Thus, the variation in the magnetization as a function of the applied magnetic field has been measured at 140 K for the H1.25-CR sample (see Figure 10). The square shape of the hysteresis loop suggests that the alignment has been improved by the dispersion of the nano-rods in the toluene solution. Indeed, a higher remanence to saturation ratio of *M_R_/M_S_* = 0.8 has been obtained (see Figure 10). Although the orientation of the wires was only partially successful, as revealed by the *M_R_/M_S_* ratio (for an ideally parallel assembly, it should be 1), the value of the coercivity is significantly high: 7.2 kOe for nano-rods free of conical heads (H1.25-CR) with an optimized aspect ratio.

These results are comparable to those from Gandha et al., [29] but, in our case, the NRs are obtained with a simple polyol synthesis technique assisted by a magnetic field. It is also important to notice that, using the theoretical model of Zeng et al. [36] that describes the coercive field of nanowires as a function of the temperature, it is possible to estimate the coercivity at room temperature of the H1.25-CR sample (see Equation (6) in [36]) presented in Figure 8. The obtained value is coherent with that measured for the corresponding NRs in an open reactor, and presented in Table 2 (i.e., ~6000 Oe for H1.25-OR).

Finally, the present work shows that the absence of conical heads at the ends of the NRs led to a significant enhancement in coercivity. However, it should be noted that the disappearance of these conical heads occurred for NRs with a very high aspect ratio, larger than 10. Thus, these two phenomena appear to be interdependent, suggesting, so far, that the improvement in *H_C_* may be correlated to both these parameters and the stacking fault density decrease.

## 4. Conclusions

Applying an external magnetic field during the synthesis of magnetic nanoparticles has led to Co nano-rods crystallizing in the hexagonal close-packed structure. It has been possible to vary the mean length and the aspect ratio of the rods by varying the strength of the external magnetic field during the synthesis. In addition, it was shown that autogenous pressure can help to increase the mean aspect ratio, to reduce the stacking fault density, as well as to decrease the conical heads. For the first time, very well-crystallized anisotropic particles with an average length of 570 nm, and a diameter in the nanometer range (15 nm), have been obtained using the polyol process. Contrary to previously reported Co nano-rods prepared without an external magnetic field, in this work, it was possible to obtain nanoparticles that do not have conical heads at their extremities, presenting a very low stacking faults density. The magnetic properties of assemblies of cobalt nano-rods have been greatly improved by applying an external magnetic field during the synthesis. The improvement in coercivity (5.8 kOe at 300 K) is strictly related to the optimization of the rods shape, their higher mean length, the absence of conical heads, and the low stacking faults density. The improvement in the *M_S_* (approaching 92% of the bulk) and of the remanence-to-saturation ratio (*M_R_/M_S_* = 0.8) is related to the fact that the longer rods obtained at the higher field tend to adopt a parallel alignment during the synthesis. The possibility to obtain such long rods opens new lines of research concerning the study of electrical conduction in metallic ferromagnetic nanowires.

## Figures and Tables

**Figure 1 nanomaterials-10-00334-f001:**
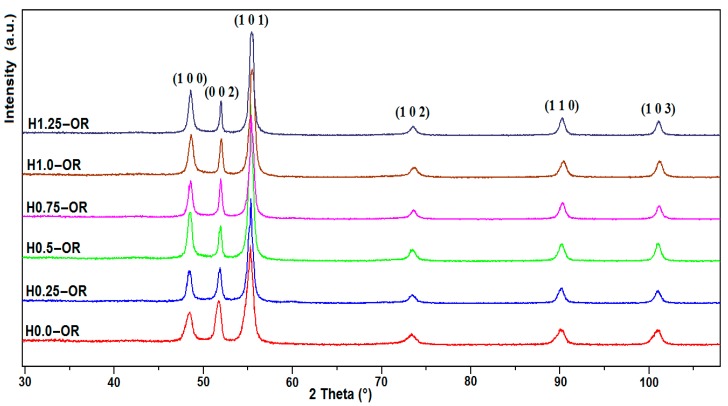
XRD pattern of Co nano-rod (NR) powders prepared by reduction of cobalt laurate in an open reactor in the presence of an applied magnetic field varying from 0 to 1.25 T.

**Figure 2 nanomaterials-10-00334-f002:**
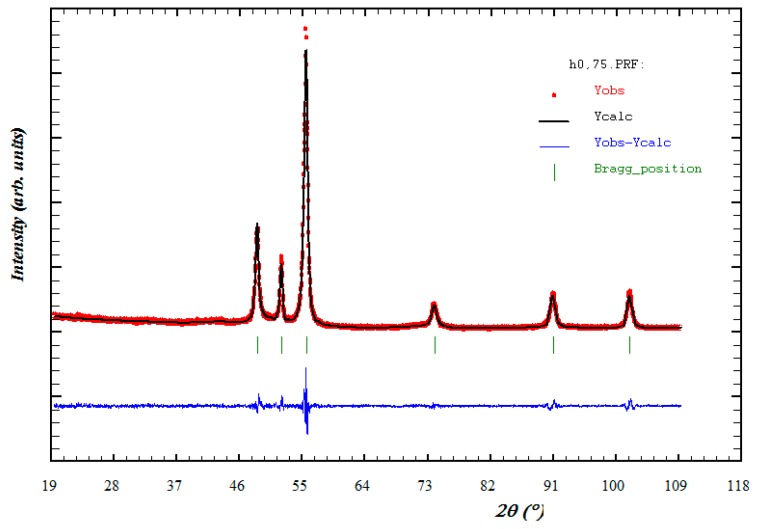
Rietveld refinement results of XRD patterns of H0.75-OR sample. The experimental data are shown as dots; the global fitting profile and the difference curve are shown as solid lines; the calculated reflection positions are indicated by stick marks.

**Figure 3 nanomaterials-10-00334-f003:**
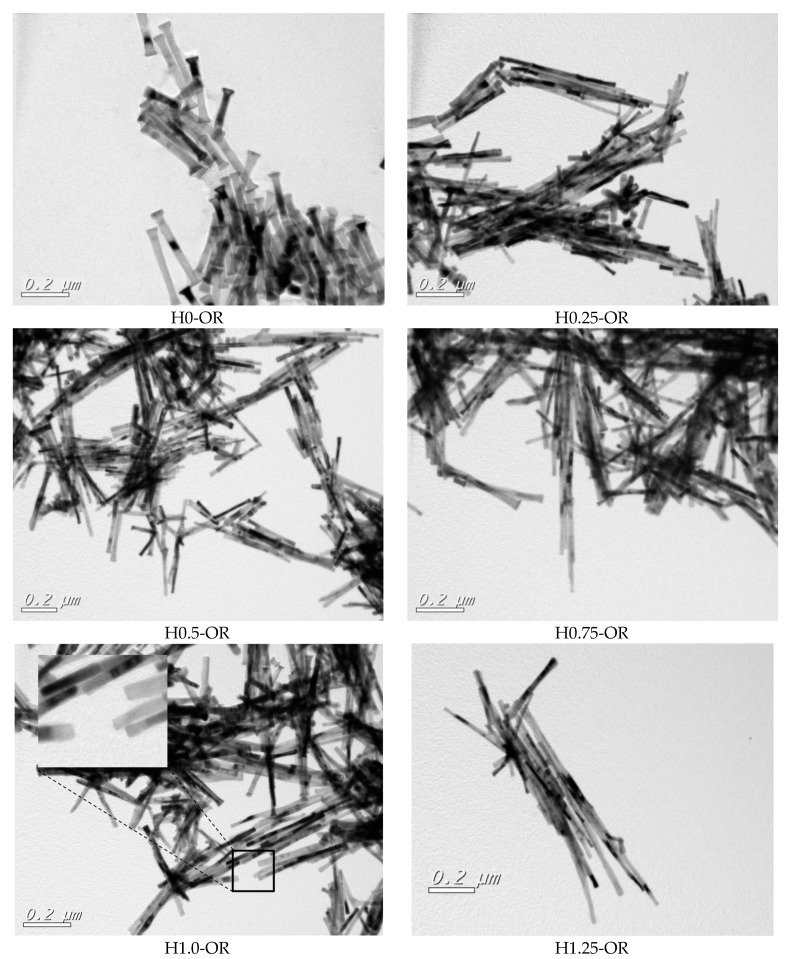
TEM images of cobalt nano-rod samples elaborated with the presence of an external magnetic field varying from 0 to 1.25 T in an open reactor. The magnified view given in the inset shows more clearly the presence/absence of the conical heads for each sample.

**Figure 4 nanomaterials-10-00334-f004:**
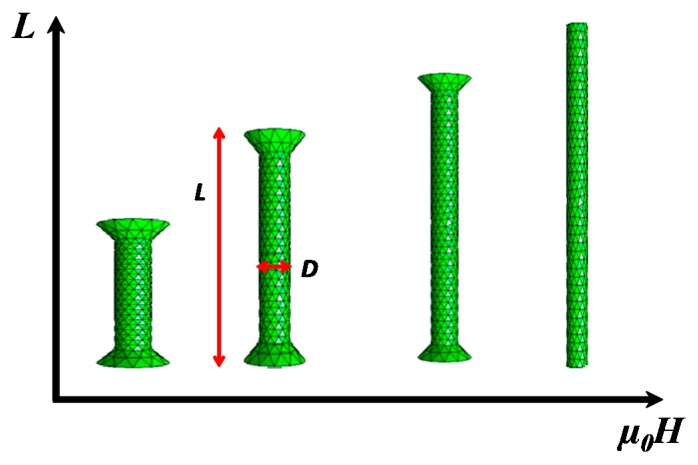
Scheme of the morphology evolution of Co nanowires as a function of the external applied magnetic field.

**Figure 5 nanomaterials-10-00334-f005:**
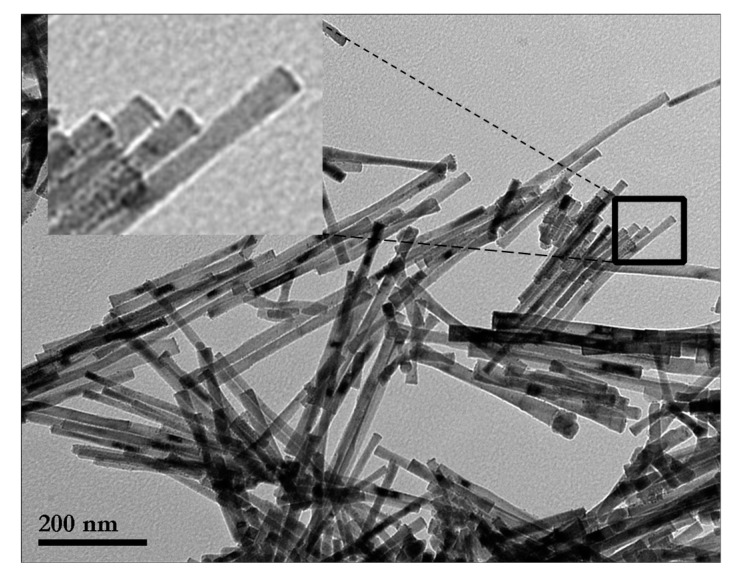
TEM of pure Co nano-rods elaborated in a closed reactor (CR) and under external magnetic field of 1.25 T.

**Figure 6 nanomaterials-10-00334-f006:**
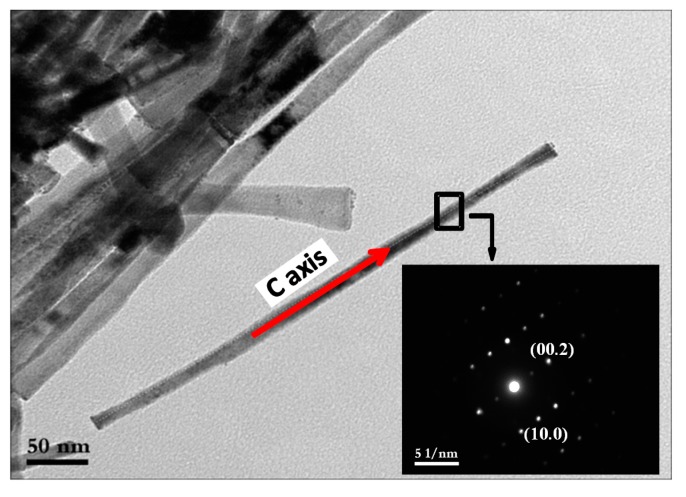
TEM image of a single Co NR (H1.25-CR) and its electron diffraction pattern corresponding to the selected zone in the TEM image.

**Figure 7 nanomaterials-10-00334-f007:**
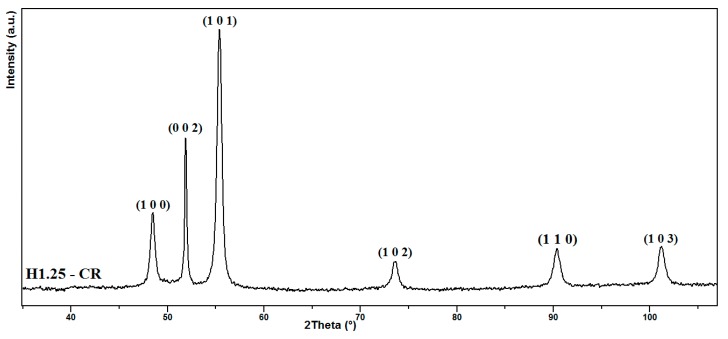
XRD pattern of Co nano-rod powders prepared by reduction of cobalt laurate in the presence of a magnetic field of 1.25 T in a closed reactor.

**Figure 8 nanomaterials-10-00334-f008:**
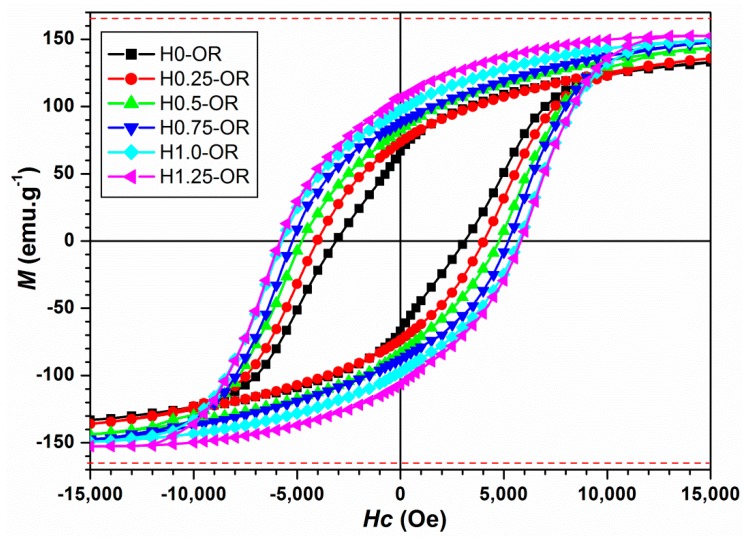
Hysteresis cycles at 300 K of cobalt nano-rods-disordered powders blocked into a resin and prepared by magnetic field-assisted polyol synthesis in an open reactor (magnetic field varied from 0 to 1.25 T). Red dashed lines represent the cobalt bulk expected value for the magnetization at saturation.

**Figure 9 nanomaterials-10-00334-f009:**
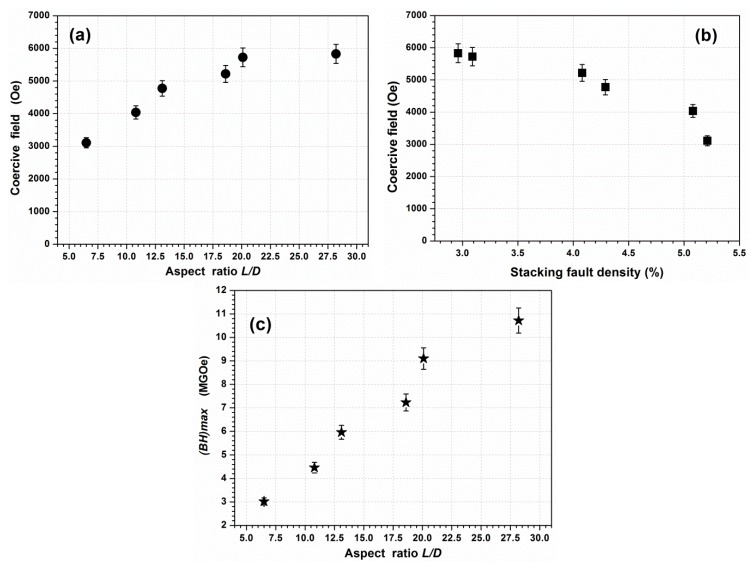
(**a**) Coercive field vs. the aspect ratio, (**b**) coercive field vs. the stacking fault density, and (**c**) (*BH*)*max* vs. the aspect ratio of Co nano-rods elaborated in an open reactor and under an applied magnetic field varying from 0 to 1.25 T.

**Figure 10 nanomaterials-10-00334-f010:**
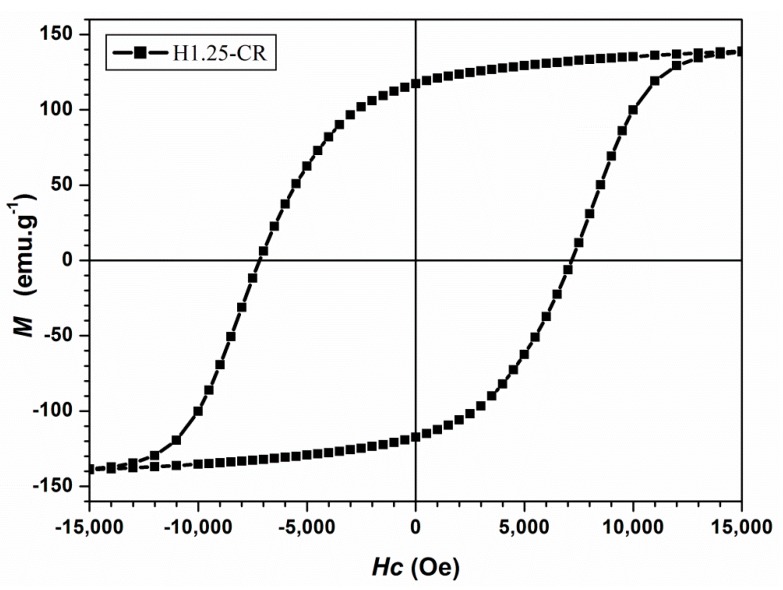
Hysteresis cycle at 140 K of Co nano-rods without conical heads (H1.25-CR).

**Table 1 nanomaterials-10-00334-t001:** Main microstructural characteristics of samples prepared in open reactor.

Samples	TEM			XRD (*L*_hkl_ ^a^), nm			*δ*^d^, %
	*L*(*σ*) ^b^, nm	*D*(*σ*) ^c^, nm	*L/D*	*L* _100_	*L* _002_	*L* _110_	
**H0-OR**	256(39)	35(13)	6.5	10.9	19.2	11.9	5.21
**H0.25-OR**	335(25)	31(11)	10.8	15.3	19.7	17.5	5.08
**H0.5-OR**	418(42)	32(7)	13.1	15.4	23.3	15.5	4.29
**H0.75-OR**	465(48)	25(5)	18.6	17.0	24.5	17.5	4.08
**H1.0-OR**	482(54)	24(4)	20.1	17.3	32.4	18.4	3.09
**H1.25-OR**	538(85)	19(5)	28.2	13.2	33.8	17.2	2.96
**H1.25-CR**	563(79)	15(4)	37.5	17.5	78.8	17.6	1.27

^a^*L*_hkl_: Diffraction coherent domain size, ^b^*L*: Mean length, *σ*: Standard deviation, ^c^*D*: Mean diameter, ^d^*δ*: Stacking fault density.

**Table 2 nanomaterials-10-00334-t002:** Static magnetic characteristics of magnetic field-assisted nano-rods elaborated in an open reactor.

Samples	*H_C_* (Oe)	*M_R_* (emu/g)	*M_S_* (emu/g)	*M_R_/M_S_*	*(BH)max* (MGOe)	Aspect Ratio (*L/D*)
**H0-OR**	3109	66.8	133.0	0.502	3.02	6.5
**H0.25-OR**	4037	73.5	136.1	0.540	4.46	10.8
**H0.5-OR**	4775	82.9	144.2	0.575	5.96	13.1
**H0.75-OR**	5218	87.9	147.7	0.595	7.23	18.6
**H1.0-OR**	5724	97.5	149.8	0.651	9.10	20.1
**H1.25-OR**	5830	106.8	152.5	0.700	10.72	28.2

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
