# Peer review of "Enhanced Magnetic Behavior of Cobalt Nano-Rods Elaborated by the Polyol Process Assisted with an External Magnetic Field"

_nanomaterials, 2020, doi:10.3390/nano10020334_

Round 1

Reviewer 1 Report

This paper shows interesting results that deserve publication in Nanomaterials. However, the authors must make the following corrections that are needed to check/verify their conclusions.

TEM studies:the microstructure of the rods  must be improved. In particular, the data presented do not show clearly the presence/absence of conical heads at the ends of the nanorods. Also the stacking faults are not presented clearly. Furthermore, why the grain size was not measured directly by TEM? Hysteresis loops: How was Ms measured? The magnetization measured at the highest field is not saturated and the law of approach to saturation should have been used to determine the real value of Ms. Also, corrections must be made for the demagnetization factor to determine the real remanence Mr. The Mr/Ms ratio should involve the new Ms and Mr values and be used in the figures. Furthermore, The real (not the normalized) M(H) loop must be shown in Figure 8. Are the loops presented obtained on randomly oriented rods? Provide more information why the longer rods align better.

Author Response

Reviewer #1: Comments and Suggestions for Authors

This paper shows interesting results that deserve publication in Nanomaterials. However, the authors must make the following corrections that are needed to check/verify their conclusions.

1- TEM studies: the microstructure of the rods must be improved. (a) In particular, the data presented do not show clearly the presence/absence of conical heads at the ends of the nanorods. (b) Also the stacking faults are not presented clearly. (c) Furthermore, why the grain size was not measured directly by TEM?

Response 1:

 (a) In each figure an insert was added to show the presence/absence of conical heads at the ends of the nanorods. A general behavior is observed: when the applied field increased, the conical heads tend to disappear. This is clearly shown when the applied field reach its higher value (Fig.2 H1.25-OR and Fig.3H1.25-CR).

(b) Several works cited in the manuscript were devoted to the stacking faults in Co nanorods and its influence on the magnetic properties of this material (Kha et al. J. Magn.Magn. Mater. 2017) (Soumare et al.,Adv. Funct. Mater. 2009) . In fact, Co exists in two crystallographic varieties: the hcp form is stable up to 450°C and transformed above into fcc one. However, each variety presents structural defects consisting in stacking faults which can be considered as an intergrowth between the two varieties. In the case of the hcp variety which corresponds to ABABAB packing sequences, few ABCABC packing sequences can be intercalated in between.  In the present work our purpose was limited to estimate the stacking faults density by X-ray diffraction analysis according to the model proposed in the several works cited.

(c) The nanorods are constituted by crystallites. The size of the crystallites was inferred from XRD analysis.  In the present work, the grains correspond in fact to the nanorods. Their sizes (length and diameter) were measured by TEM by counting at least 200 grains (nanorods) in each case. Both series are presented in table 1 and compared.

2- Hysteresis loops: How was Ms measured? The magnetization measured at the highest field is not saturated and the law of approach to saturation should have been used to determine the real value of Ms.

Response 2:

It is true that for the sake of clarity on the hysteresis shape and coercivity value, we decided to present in figure 7 and 8 the magnetization behavior up to 15000 Oe which is a zoom of the complete hysteresis cycle experimentally measured. This prevents the reviewer to correctly verify the saturation behavior.  

Figure 1 below shows an example of a hysteresis cycle of cobalt nano-rods (H1.25-OR) from -2T to 2T. It is clear that the hysteresis cycle is saturated at the highest field (H > 1.5 Tesla) as reported into the text.  The value of Ms is deduced by linear extrapolation.

On the both part of the hysteresis cycle (half-positive and half-negative) the found value of Ms is the same within the experimental error, thus proving also a proof of the symmetry of the Hysteresis loop.

Figure 1: Hysteresis cycle at 300 K of cobalt nano-rods (H1.25-OR) prepared under an external magnetic field of 1.25 T in an open reactor.

In order to persuade the reviewer that our extrapolation is in great agreement with the law of approach to saturation, we performed the suggested fit of the experimental data at high magnetic field(Figure 2). We could then compare the Ms values obtained by this latter approach to the Ms values obtained by the linear extrapolation proposed. The highest difference obtained by the comparison (see Table 1 below) is less than 0.8 emu/g.

Figure 2: Red lines represent the law of approach to saturation for cobalt nano-rods prepared in the open reactor under different magnetic field applied.

Table 1: Values of Ms found by linear extrapolation and by the law of approach to saturation for cobalt nano-rods prepared in open reactor under an external magnetic field (0 T up to 1.25 T).

Samples

Ms by linear extrapolation (emu/g)

Ms by law of approach to saturation (emu/g)

H0-OR

133.0

132.8

H0.25-OR

136.1

136.8

H0.5-OR

144.2

144.7

H0.75-OR

147.7

148.5

H1.0-OR

149.8

149.6

H1.25-OR

152.5

153.2

3- Also, corrections must be made for the demagnetization factor to determine the real remanence Mr. The Mr/Ms ratio should involve the new Ms and Mr values and be used in the figures.

Response 3:

The geometric factor of the measured powders is already taken into account during the measurement (VSM or SQUID). It is important to underline that we don't measure a single nanowire, a thin layer or a bulk sample with a well-defined macroscopic geometry. In our case, the powder is distributed isotropically in space and the total magnetic volume is very small compared to the scanned volume by the DC measurements. Thus the demagnetizing factor in our case is very close to 1. The real remanence can be thus calculated by Mr = M(H = 0kOe) and the Mr/Ms ratio doesn't change in Table 2.

4- Furthermore, The real (not the normalized) M(H) loop must be shown in Figure 8.

Response 4 :

According to the reviewer suggestion the real M(H) loop is shown in the revised version of the article (see Figure 10 which corresponds to the fig.8 in our previous version).

5- (a) Are the loops presented obtained on randomly oriented rods? (b) Provide more information why the longer rods align better.

Response:

In Figure 9, the loop corresponds to randomly oriented nanorods as described in the manuscript. In Figure 10 the hysteresis cycle corresponds to the oriented powder in toluene before the performed cycle at 140K.  The powder was dispersed them in liquid toluene and applied a magnetic field of 50 kOe during the freezing of the solution (at 140 K). Following the suggestion of the reviewer, we have improved the new version of the article with a more detailed explanation on why the longer rods align better. In the revised version, the influence of the presence/absence of conical ends is also discussed in order to explain the better alignment observed for longer nanorods which present at the same time a better cylindrical morphology (i.e. without conical heads at the end of the nanorods) .

Added part (lines: 309-324)

“As shown in Table 2, the ratio Mr/Ms increases with the aspect ratio L/D indicating that the NRs align parallel to each other spontaneously better when their length (see Table 1) increases. As reported previously [34, 35] the optimized shape anisotropy (L/D=28) induces an easy magnetization axis along the length of the NRs inducing, at its turn, a high dipolar field outside of the NR. This field is radial around the NR ends and parallel along the lateral surface and decreases quickly as function of the distance. The dipolar interaction of this magnetic field with the magnetic moment of the nearby NRs, tends to align close NRs parallel to each other. This is more and more effective when the magnetic moment is maximized inside the NRs and the magnetic anisotropy force it to align along the length. This is the case in the optimized aspect ratio NRs (see Table 1 and 2). It has also been demonstrated [9-11] that the radial dipolar field at the end of the NRs strongly depends on the shape of the NRs heads. Conical head at the end of the NR creates higher demagnetizing field outside this latter, compared to the one at the head of the cylindrical NR. This has a catastrophic effect on the spontaneous alignment of the nearby NRs as well as on their coercivity value. As reported before [35] the magnetization reversal mechanism of NRs ensemble strongly depends on the orientation of few NRs close to the head of the nearby ones. Our measurements in Figure 8 are in full agreement with those previous observations.

Reviewer #1: Comments and Suggestions for Authors

This paper shows interesting results that deserve publication in Nanomaterials. However, the authors must make the following corrections that are needed to check/verify their conclusions.

1- TEM studies: the microstructure of the rods must be improved. (a) In particular, the data presented do not show clearly the presence/absence of conical heads at the ends of the nanorods. (b) Also the stacking faults are not presented clearly. (c) Furthermore, why the grain size was not measured directly by TEM?

Response 1:

 (a) In each figure an insert was added to show the presence/absence of conical heads at the ends of the nanorods. A general behavior is observed: when the applied field increased, the conical heads tend to disappear. This is clearly shown when the applied field reach its higher value (Fig.2 H1.25-OR and Fig.3H1.25-CR).

(b) Several works cited in the manuscript were devoted to the stacking faults in Co nanorods and its influence on the magnetic properties of this material (Kha et al. J. Magn.Magn. Mater. 2017) (Soumare et al.,Adv. Funct. Mater. 2009) . In fact, Co exists in two crystallographic varieties: the hcp form is stable up to 450°C and transformed above into fcc one. However, each variety presents structural defects consisting in stacking faults which can be considered as an intergrowth between the two varieties. In the case of the hcp variety which corresponds to ABABAB packing sequences, few ABCABC packing sequences can be intercalated in between.  In the present work our purpose was limited to estimate the stacking faults density by X-ray diffraction analysis according to the model proposed in the several works cited.

(c) The nanorods are constituted by crystallites. The size of the crystallites was inferred from XRD analysis.  In the present work, the grains correspond in fact to the nanorods. Their sizes (length and diameter) were measured by TEM by counting at least 200 grains (nanorods) in each case. Both series are presented in table 1 and compared.

2- Hysteresis loops: How was Ms measured? The magnetization measured at the highest field is not saturated and the law of approach to saturation should have been used to determine the real value of Ms.

Response 2:

It is true that for the sake of clarity on the hysteresis shape and coercivity value, we decided to present in figure 7 and 8 the magnetization behavior up to 15000 Oe which is a zoom of the complete hysteresis cycle experimentally measured. This prevents the reviewer to correctly verify the saturation behavior.  

Figure 1 below shows an example of a hysteresis cycle of cobalt nano-rods (H1.25-OR) from -2T to 2T. It is clear that the hysteresis cycle is saturated at the highest field (H > 1.5 Tesla) as reported into the text.  The value of Ms is deduced by linear extrapolation.

On the both part of the hysteresis cycle (half-positive and half-negative) the found value of Ms is the same within the experimental error, thus proving also a proof of the symmetry of the Hysteresis loop.

Figure 1: Hysteresis cycle at 300 K of cobalt nano-rods (H1.25-OR) prepared under an external magnetic field of 1.25 T in an open reactor.

In order to persuade the reviewer that our extrapolation is in great agreement with the law of approach to saturation, we performed the suggested fit of the experimental data at high magnetic field(Figure 2). We could then compare the Ms values obtained by this latter approach to the Ms values obtained by the linear extrapolation proposed. The highest difference obtained by the comparison (see Table 1 below) is less than 0.8 emu/g.

Figure 2: Red lines represent the law of approach to saturation for cobalt nano-rods prepared in the open reactor under different magnetic field applied.

Table 1: Values of Ms found by linear extrapolation and by the law of approach to saturation for cobalt nano-rods prepared in open reactor under an external magnetic field (0 T up to 1.25 T).

Samples

Ms by linear extrapolation (emu/g)

Ms by law of approach to saturation (emu/g)

H0-OR

133.0

132.8

H0.25-OR

136.1

136.8

H0.5-OR

144.2

144.7

H0.75-OR

147.7

148.5

H1.0-OR

149.8

149.6

H1.25-OR

152.5

153.2

3- Also, corrections must be made for the demagnetization factor to determine the real remanence Mr. The Mr/Ms ratio should involve the new Ms and Mr values and be used in the figures.

Response 3:

The geometric factor of the measured powders is already taken into account during the measurement (VSM or SQUID). It is important to underline that we don't measure a single nanowire, a thin layer or a bulk sample with a well-defined macroscopic geometry. In our case, the powder is distributed isotropically in space and the total magnetic volume is very small compared to the scanned volume by the DC measurements. Thus the demagnetizing factor in our case is very close to 1. The real remanence can be thus calculated by Mr = M(H = 0kOe) and the Mr/Ms ratio doesn't change in Table 2.

4- Furthermore, The real (not the normalized) M(H) loop must be shown in Figure 8.

Response 4 :

According to the reviewer suggestion the real M(H) loop is shown in the revised version of the article (see Figure 10 which corresponds to the fig.8 in our previous version).

5- (a) Are the loops presented obtained on randomly oriented rods? (b) Provide more information why the longer rods align better.

Response:

In Figure 9, the loop corresponds to randomly oriented nanorods as described in the manuscript. In Figure 10 the hysteresis cycle corresponds to the oriented powder in toluene before the performed cycle at 140K.  The powder was dispersed them in liquid toluene and applied a magnetic field of 50 kOe during the freezing of the solution (at 140 K). Following the suggestion of the reviewer, we have improved the new version of the article with a more detailed explanation on why the longer rods align better. In the revised version, the influence of the presence/absence of conical ends is also discussed in order to explain the better alignment observed for longer nanorods which present at the same time a better cylindrical morphology (i.e. without conical heads at the end of the nanorods) .

Added part (lines: 309-324)

“As shown in Table 2, the ratio Mr/Ms increases with the aspect ratio L/D indicating that the NRs align parallel to each other spontaneously better when their length (see Table 1) increases. As reported previously [34, 35] the optimized shape anisotropy (L/D=28) induces an easy magnetization axis along the length of the NRs inducing, at its turn, a high dipolar field outside of the NR. This field is radial around the NR ends and parallel along the lateral surface and decreases quickly as function of the distance. The dipolar interaction of this magnetic field with the magnetic moment of the nearby NRs, tends to align close NRs parallel to each other. This is more and more effective when the magnetic moment is maximized inside the NRs and the magnetic anisotropy force it to align along the length. This is the case in the optimized aspect ratio NRs (see Table 1 and 2). It has also been demonstrated [9-11] that the radial dipolar field at the end of the NRs strongly depends on the shape of the NRs heads. Conical head at the end of the NR creates higher demagnetizing field outside this latter, compared to the one at the head of the cylindrical NR. This has a catastrophic effect on the spontaneous alignment of the nearby NRs as well as on their coercivity value. As reported before [35] the magnetization reversal mechanism of NRs ensemble strongly depends on the orientation of few NRs close to the head of the nearby ones. Our measurements in Figure 8 are in full agreement with those previous observations.

Reviewer 2 Report

Please consider the following:

1. The introduction mentions the potential usability of optimized high aspect ratio NRs in "areas such as measuring the properties of electrical conduction in metallic nanowires". A similar statement is made in Conclusions. I believe that some details and references would be welcome.

2. Add a more detailed description of the experimental setup for NR preparation - a schematic drawing of the setup and of the preparation steps would be helpful. Please offer more details regarding the addition of the (relatively strong) magnetic field. Also, some details and discussions about the use of open reactor / closed reactor would be necessary. Can you provide quantitative indication of the pressure difference between the two cases? (If overall length is an issue, consider adding a supplementary material.)

3. I find the term "conical tip" somehow inadequate, as "tip" suggests the presence of a sharp structure at the extremity of the NRs, which is not the case here. For the description of your NR terminations I would suggest something like "truncated cone termination" - again, a schematic drawing of their geometry would be helpful, such as those in references 7-9.

4. Figures:

The insets in Figs. 2 and 4 are very small and the letters are hardly legible. Is the inset in the last image of Fig. 2 significant? If not, it should be removed. If yes, it should be discussed in the text.

In the inset of Fig. 4 we find (00.1). Shouldn't this be (00.2)?

In the graph of Fig. 7a the connecting line appears to suggest a slight decrease of Hc at higher L/D values, but the text refers to saturation of Hc. I suggest removing the line and keeping only the data points.

5. Some considerations regarding grammar and syntax (not exhaustive):

use "open reactor" instead of "opened reactor" throughout the text

lines 215 and 216: "the role of autogenous pressure since in the cited work remains to be discussed" - the word "since" appears to be unnecessary

line 223: replace "are" with "is"

line 227: consider rephrasing "without mentioning their alignments"

line 230: replace ";" with ","

line 236: remove "with"

table 2, header: for "emu.g−1" please use the proper multiplication dot, and superscript for the exponent; or use "/g"; alternatively, you could consider using SI units (A·m2/kg, or Wb·m/kg) with careful conversion if necessary

line 275: replace "than" with "as"

line 276: replace "have" with "has"

line 301: replace "yielded "with "led"

line 309: add "density" after "defects"

line 314: replace "are" with "is"; replace "tends" with "tend"

Good luck! :)

Author Response

Reviewer #2: Comments and Suggestions for Authors

Please consider the following:

The introduction mentions the potential usability of optimized high aspect ratio NRs in "areas such as measuring the properties of electrical conduction in metallic nanowires". A similar statement is made in Conclusions. I believe that some details and references would be welcome.

Response 1:

In the literature, the more studied systems are based on noble metals (Ag, Au). For non-noble metals, only Ni has been studied. Co Nanorods with very long length elaborated in the present study, offer a new field of investigations for inter-connected unidimensional systems that can respond to an external magnetic field stimulus. We add two references on the revisited version in order to reinforce our statements (Nirmalraj et al. Nano Letters 12, 2012) (Langley et al. Nanoscale, 6, 2012)

According to the reviewer suggestion, the following details were added in the revised version (lines 52-55:

Indeed, the main studied systems are based on noble metals nanowires (Ag, Au)   [Nirmalraj et al. Nano Letters 12, 2012)][Langey et al. Nanoscale 6, 2014]. Co Nanorods with very long length elaborated in the present study, offer so far a new field of investigations for interconnected unidimensional systems that can respond to an external magnetic field stimulus.

Add a more detailed description of the experimental setup for NR preparation - a schematic drawing of the setup and of the preparation steps would be helpful.

Please offer more details regarding the addition of the (relatively strong) magnetic field.

Also, some details and discussions about the use of open reactor / closed reactor would be necessary. Can you provide quantitative indication of the pressure difference between the two cases? (If overall length is an issue, consider adding a supplementary material.)

Response 2:

According to the reviewer suggestion, we have added in the revised version a more detailed description of the experimental setup taking into account all the remarks (application of magnetic field, quantitative indication of the pressure difference between the open reactor  and closed one).

Detailed description added in the revised version (lines 102-113):

“The synthesis was conducted in two reactor types having the same capacity (70 mL). The first one is an open double-walled glass reactor equipped with a cooler. The second one is a closed double-walled made of non-magnetic austenitic steel. This latter is hermetically sealed with a threaded plug to maintain the pressure. In both cases, the heating is ensured by the circulation in the space surrounding the inner wall of heat transfer oil using a thermostatically controlled bath (see Fig.1 in Supplement material).

The reactor with the mixture introduced in the inner layer, is placed in the air gap of the electromagnet (See Fig. 1 in supplement material). The oil bath is then heated to 170 °C with a temperature ramping rate of 8 °C/min. Thanks to a prior magnetic field calibration (See Fig. 2 in supplementary material, it is possible to vary the applied magnetic field from 0 to 1.25 T by controlling the value of the current applied to the electromagnet. The magnetic field is maintained during both synthesis and cooling steps”.

Also the pressure in both cases was indicated in the revised version (1 atm for OR and 50 atm for CR) (line 115, line 120)

Furthermore, schematic drawing of the setup and of the preparation steps is given in Figure S1 of supplementary material.

I find the term "conical tip" somehow inadequate, as "tip" suggests the presence of a sharp structure at the extremity of the NRs, which is not the case here. For the description of your NR terminations I would suggest something like "truncated cone termination"

- again, a schematic drawing of their geometry would be helpful, such as those in references 7-9.

Response 3:

We agree with the reviewer. In our case, the use of the term "conical tip" is inadequate. According to Ott F. et al. (reference 7 in previous version), we have used the term "conical edge or conical end". The corrections are made in the revised version all along the main text (the terms used are conical ends or conical heads).

Also the reviewer question let us understand the importance to be clearer on the shape evolution of the NRs.  Seeking a better understanding, we added a schematic drawing of nanowires geometry in the revised version (see figure below which is the Figure 4 of the resubmitted work).

Figure 4: Schematic evolution of the Co nanowires morphology as a function of the external applied field during the synthesis.

Figures:

The insets in Figs. 2 and 4 are very small and the letters are hardly legible. Is the inset in the last image of Fig. 2 significant? If not, it should be removed. If yes, it should be discussed in the text.

Response 4: The inset in Fig.2 is not significant. Thus it was removed according the reviewer suggestion. In the revised version this Figure becomes Figure 3

The inset in Fig.4 has been changed taking into account the reviewer suggestions. Thus more legible letters are used and (00.1) has been changed into (00.2). This figure is now Figure 6.

In the inset of Fig. 4 we find (00.1). Shouldn't this be (00.2)?

Response: Yes it corresponds in fact to (00.2) indexation. This has been changed

In the graph of Fig. 7a (Figure 9a in the revised version) the connecting line appears to suggest a slight decrease of Hc at higher L/D values, but the text refers to saturation of Hc. I suggest removing the line and keeping only the data points.

Response: Line has been removed

Some considerations regarding grammar and syntax (not exhaustive):

use "open reactor" instead of "opened reactor" throughout the text

Opened  replaced by Open

lines 215 and 216: "the role of autogenous pressure since in the cited work remains to be discussed" - the word "since" appears to be unnecessary: “since “ was removed

line 223: replace "are" with "is" Done

line 227: consider rephrasing "without mentioning their alignments"

"without mentioning their alignments" is replaced by "and their magnetic alignments"

line 230: replace ";" with "," Done

line 236: remove "with" Done

table 2, header: for "emu.g−1" please use the proper multiplication dot, and superscript for the exponent; or use "/g"; alternatively, you could consider using SI units (A·m2/kg, or Wb·m/kg) with careful conversion if necessary Done

line 275: replace "than" with "as" Done

line 276: replace "have" with "has" Done

line 301: replace "yielded "with "led" Done

line 309: add "density" after "defects" Done

line 314: replace "are" with "is"; replace "tends" with "tend" Done

Reviewer 3 Report

The paper reports on the formation and characterization of Co nanorods using a polyol synthesis procedure assisted by the application of a magnetic field. It is observed that enhanced coercivity is obtained on such Co nanorods compared with counterparts obtained with similar procedure but without applying magnetic field. The paper is well written and presents sound results. 

Several issues have to be addressed before considering the paper for publication in Nanomaterials:

a) Regarding the XRD results: the authors do not elaborate how are they estimating the diffraction coherent domain size and what program was used for fitting the XR diffractograms. Also, it is not clear why the authors do not provide a full-profile analysis and calculate the mean grain size from the Lhkl values, averaged over all hkl reflections, in order to compare averaged results with TEM results on the domain size.

b) The authors comment that improved magnetic properties in samples prepared under applied magnetic field are mostly due to the absence of the formed conical tips at the end of rods. This is unlikely to be the most influential factor. It is known that the applied magnetic field for instance during colloidal crystallization of Co or Co-containing nanoparticles has a strong effect of aligning Co atoms that have strong magnetic moments and it is much more probable that the field assisted Co atoms alignment over longer distances creates better magnetic properties. That is why the authors observe longer formed nanorods with better aspect ratio and simply due to the longer (in average) nanorods they observe better coercivity in the samples synthesized under applied field.

c) It would be useful to plot also the values of maximum energy product (BH)max as a function of the aspect ratio of the nanorods and also as a function of the applied magnetic field, since (BH)max represents the energy stored by the magnetic system and is one of the main parameters in estimating the good properties of a hard magnetic system.

Provided these issue are conveniently addressed the paper can be published in Nanomaterials.

Author Response

Reviewer #3: Comments and Suggestions for Authors

The paper reports on the formation and characterization of Co nanorods using a polyol synthesis procedure assisted by the application of a magnetic field. It is observed that enhanced coercivity is obtained on such Co nanorods compared with counterparts obtained with similar procedure but without applying magnetic field. The paper is well written and presents sound results.

Several issues have to be addressed before considering the paper for publication in Nanomaterials:

a) Regarding the XRD results: the authors do not elaborate how are they estimating the diffraction coherent domain size and what program was used for fitting the XR diffractograms. Also, it is not clear why the authors do not provide a full-profile analysis and calculate the mean grain size from the Lhkl values, averaged over all hkl reflections, in order to compare averaged results with TEM results on the domain size.

Response a:

We thank the reviewer for his/her remarks. In the previous version, we have used the High Score Plus software for fitting the XR diffractograms. Indeed, the coherent domain size diffraction was calculated by Debye-Scherrer formula (see lines 110-111). The mean crystallite size determined from the hkl reflections are given in Table 1 of the main text in the revised version. We have also compared the crystallite size inferred from X-ray diffraction with ones determined from the TEM observation (see lines 146-148).

In the revised version, to go further in our study, we have used a Rietveld refinement of XRD patterns. This study allowed determining with more precision the coherent domain size diffraction.

The corresponding changes in the revised are (lines 129-130):

“The lattice parameters and the size of coherent diffraction domains (crystallites),Lhkl, were calculated using the FullProf program [13] based on the Rietveld method [14]

A Rietveld refinement model is presented in Figure 2 of the revised version and modifications have been made in Table 1 (all the Rietveld refinements of our powders are given in supplementary material S2). In fact, the results found by Rietveld refinements are similar to that calculated by Scherrer method. As Rietveld method is more reliable, we have chosen to replace the values of mean crystallite size calculated by Scherrer method by the ones obtained by Rietveld refinements.

Figure 2. Rietveld refinement results of XRD patterns of H0.75-OR sample. The experimental data are shown as dots; the global fitting profile and the difference curve are shown as solid lines; the calculated reflection positions are indicated by stick marks.

Table 1. Main microstructural characteristics of samples prepared in open reactor.

Samples

TEM

XRD (Lhkl a ), nm

δ d, %

L(σ) b, nm

D(σ) c, nm

L/D

L100

L002

L110

H0-OR

256(39)

35(13)

6.5

10.9

19.2

11.9

5.21

H0.25-OR

335(25)

31(11)

10.8

15.3

19.7

17.5

5.08

H0.5-OR

418(42)

32(7)

13.1

15.4

23.3

15.5

4.29

H0.75-OR

465(48)

25(5)

18.6

17.0

24.5

17.5

4.08

H1.0-OR

482(54)

24(4)

20.1

17.3

32.4

18.4

3.09

H1.25-OR

538(85)

19(5)

28.2

13.2

33.8

17.2

2.96

H1.25-CR

563(79)

15(4)

37.5

17.5

78.8

17.6

1.27

a Lhkl: diffraction coherent domain size, b L: mean length, σ: standard deviation,

 c D: mean diameter, d δ: stacking fault density.

b) The authors comment that improved magnetic properties in samples prepared under applied magnetic field are mostly due to the absence of the formed conical tips at the end of rods. This is unlikely to be the most influential factor. It is known that the applied magnetic field for instance during colloidal crystallization of Co or Co-containing nanoparticles has a strong effect of aligning Co atoms that have strong magnetic moments and it is much more probable that the field assisted Co atoms alignment over longer distances creates better magnetic properties. That is why the authors observe longer formed nanorods with better aspect ratio and simply due to the longer (in average) nanorods they observe better coercivity in the samples synthesized under applied field.

Response b:

We definitely agree with the reviewer. In fact, the formation of long nanorods can be due to the interaction of Co atoms with the applied field. This field helps Co atoms to be aligned along a privileged direction given by the magnetic field lines.

As suggested by the reviewer we discuss this point in the revised version of our work and references supporting this suggestion have been added (lines 234-239).

“Indeed, the applied field helps Co atoms, with high magnetic moment, to deposit along the c axis of the nanorods which is their easy magnetization direction supposed to be aligned along the applied field direction. This leads to the formation of longer nanorods with better crystallinity (i.e. less stacking faults) and thus better magnetic properties. Such influence has already been reported in several previous works on the synthesis of Co anistropic particles [24-26].

c) It would be useful to plot also the values of maximum energy product (BH)max as a function of the aspect ratio of the nanorods and also as a function of the applied magnetic field, since (BH)max represents the energy stored by the magnetic system and is one of the main parameters in estimating the good properties of a hard magnetic system.

Response c:

As suggested by the reviewer, (BH)max values were determined by using a previous reported calculation (Anagnostopoulou, E. ; Grindi, B. ; Lacroix, L.–M. ; Ott, F. ; Panagiotopoulos, I. ; Viau, G. Dense arrays of cobalt nanorods as rare-earth free permanent magnets, Nanoscale, 2016, 8, 4020 – 4029; HTT Nong, K Mrad, F Schoenstein, JY Piquemal, N Jouini, B Leridon, S Mercone, Advances in Natural Sciences: Nanoscience and Nanotechnology 8 (2), 025012) and reported in  Table 2 of the revisited version. The obtained values of (BH)max have been plotted as function of the aspect ratio (see Figure 9 c). Also these results were discussed in the revisited version (lines 288-295) .  

« The energy product (BH)max has been calculated on the basis of previous reported method [30, 31]. The calculations were made considering the density of pure cobalt (d = 8.92 g/cm3) and also based on a 100 percent volume fraction. The magnetic energy values obtained for the different powders also increase when the aspect ratio increases (Figure 9c) and reach 10.7 MGOe which is of the same order of magnitude than the energy reported by Ener et al. for consolidated Co nanowires [32]. This energy product can be improved with higher quality of alignment of the nanowires and their dispersion as reported by Gandha et al. [28], Anagnostopoulou et al. [30] and Fang et al. [33] ».

Reviewer 4 Report

Therefore; there is nowadays a great preoccupation worldwide the synthesis of some nanomaterials. Several research groups worldwide are currently involved in obtaining magnetic materials. The audience of the journal will definitely find it interesting to read about the formation of Co nano-rods using polyol process and the application of an external magnetic field with good hard magnetic properties at room temperature.The title is well chosen, in accordance with the content of the article. The abstract is well structured, pointing out the goal of the research, as well as the methods used in order to achieve it. The main results are briefly described and the core conclusion is presented. The abstract is self-explanatory and totally reflects the content of the article. The introduction presents the background of the study and reveals the need for a more detailed research. The aims and objectives are clearly stated in the introduction. The experiments are well designed and executed, with a high degree of reproducibility. The Materials and Methods section is very well organized, with subheadings arranged in a logical order, which describe the methodology in great detail. The methods are well chosen and represent adequate ways of achieving the main goal. The methods taken from the literature are properly cited in the text. The equipment used is adequate and well described, including the manufacturer. All experiments can easily be understood and are reproducible. The results are clearly presented, in a concise manner. Discussions are well conducted and presented in a logical order. They are supported by an adequate amount of literature data and also enlighten the originality of the study. The discussion section clearly states this discovery and therefore points out the added value of this research. Conclusions are well presented in the final paragraph of the manuscript and describe the advances that were made as an outcome of the present paper, together with the perspectives for future research in the field. The added value brought by this paper is of great importance, as it represents the starting point for future studies.

Author Response

We sincerely thank the reviewer for this positive appreciation of our work.

Round 2

Reviewer 1 Report

How was the saturation magnetization determined? Is the value reported the magnetization at 25 kOe or is it extrapolated to larger fields because the loops are not saturated?  this is very important when calculating the Mr/Ms ratio.

Author Response

Dear Reviewer,

All cycles are saturated at the highest field (high than 1.5 Tesla). The Ms values were determined by two methods: linear extrapolation and also by the law of approach to saturation. The two methods are developped in the attached file. As you can see, for each cycle, the two obtained values are very close to each other, the difference being less than 0.8 emu/g.

The manuscript has been revised and include now such precisions (lines 270-274)

"We measured, by a standard magnetometer technique, the magnetization of the blocked nano-powders as function of the applied magnetic field (hysteresis loops) at room temperature (300 K) from -2T to +2T.  The hysteresis cycles saturated at the highest fields (H >1.5T). The saturation magnetization was determined by two different methods: the linear extrapolation and the law of approach to saturation. The two obtained values are very similar, the difference being less than 0.8emu/g." 

Reviewer 3 Report

The reviewed version has included the required modifications and therefore I consider now the manuscript acceptable for publication.

Author Response

Thank you for your decision